# Are neighbourhoods of tuberculosis cases a high-risk population for active intervention? A protocol for tuberculosis active case finding

Bachti Alisjahbana[1,2], Raspati Cundarani Koesoemadinata[1,3]*, Panji Fortuna Hadisoemarto[1,4], Bony Wiem Lestari[1,3,4], Sri Hartati[1], Lidya Chaidir[1,5], Chuan-Chin Huang[6], Megan Murray[6], Philip Campbell Hill[7], Susan Margaret McAllister[7]

1 Research Center for Care and Control of Infectious Disease, Universitas Padjadjaran, Bandung, Indonesia, 2 Department of Internal Medicine, Faculty of Medicine Universitas Padjadjaran/Dr Hasan Sadikin General Hospital, Bandung, Indonesia, 3 Department of Internal Medicine, Radboud Institute for Health Sciences, Radboud University Medical Center, Nijmegen, The Netherlands, 4 Department of Public Health, Faculty of Medicine Universitas Padjadjaran, Bandung, Indonesia, 5 Division of Microbiology, Department of Biomedical Science, Faculty of Medicine Universitas Padjadjaran, Bandung, Indonesia, 6 Department of Global Health and Social Medicine, Harvard Medical School, Boston, Massachusetts, United States of America, 7 Centre for International Health, Department of Preventive and Social Medicine, University of Otago Medical School, Dunedin, New Zealand

* r.c.koesoemadinata@unpad.ac.id

**Funding:** Funding for this research comes from: 1. The e-Asia Joint Research Program administered through the Health Research Council of New

## Abstract

### Background

Indonesia has the second largest tuberculosis (TB) burden globally. Attempts to scale-up TB control efforts have focused on TB households. However, in most high burden settings, considerable *Mycobacterium tuberculosis (Mtb)* transmission occurs outside TB households. A better understanding of transmission dynamics in an urban setting in Indonesia will be crucial for the TB Control Program in scaling up efforts towards elimination of TB in a more targeted way. Therefore, the study aims to measure TB prevalence and incidence in household contacts and neighbourhoods in the vicinity of known TB cases and to assess their genomic and epidemiological relatedness.

### Methods and analysis

Individuals (~1000) living in the same household as a case diagnosed with pulmonary TB (n = 250) or in a neighbouring household (~4500 individuals) will be screened for TB symptoms and by chest x-ray. Two sputum samples will be collected for microbiological analysis from anyone with a productive cough. Any person found to have TB will be treated by the National TB Control Program. All those with no evidence of TB disease will have a repeat screen at 12 months. Whole-genome sequencing (WGS) and social network analysis (SNA) will be conducted on Index cases and contacts diagnosed with TB.

Zealand, Contract No. 19/897 (award receiver: SMM). URL: https://www.the-easia.org/ 2. Indonesian Research and Technology Grant/ National Research and Innovation Agency administered through Universitas Padjadjaran, Contract No 1827/UN6.3.1/LT/2020 (award receiver: BA). URL: https://www.brin.go.id/ The funders had and will not have a role in study design, data collection and analysis, decision to publish, or preparation of the manuscript.

**Competing interests:** The authors have declared that no competing interests exist.

# Introduction

Tuberculosis (TB) remains a leading cause of infection globally, with an estimated 10 million new cases of disease and 1.2 million deaths from TB in 2019 [1]. To eliminate TB, we need to identify target populations most at risk so that enhanced intervention can drive the incidence of TB down across the wider population. Many attempts to intensify TB control efforts have focused on TB households [2–4]. However, we now know that in most high burden settings, the majority of *Mycobacterium tuberculosis (Mtb)* transmission occurs in the community where there are vastly greater numbers of human encounters [5] that often share socio-economic risk factors that are correlated with TB [6, 7]. In a feasibility study in Bandung, Indonesia [8] we followed individual TB patients back to their communities. We found an incidence rate of self-reported TB of 649 per 100,000/year in TB household contacts, 675 in neighbouring households, and 325 in randomly selected neighbourhoods [8].

TB disease occurs through two distinct mechanisms (primary progression and reactivation) that require different interventions. Soon after exposure to *Mtb*, a small proportion of people develop TB disease (primary progression) and approximately 50% of them are latently infected but remain well. Of these latently infected people, 5% develop TB disease during their lifetime (reactivation) [9–11]. Early diagnosis and six months of multi-drug treatment of TB disease reduces transmission to case contacts, and preventive treatment of latent TB for as little as three months reduces reactivation in infected contacts by up to 90% [12, 13]. Both strategies are essential for TB elimination [14]. If neighbourhoods around TB cases have high rates of TB, then this is likely to be due to a particular mix of primary progression and reactivation disease.

Indonesia has the second largest TB burden globally (prevalence estimate 647/100,000 population) [15]. Bandung City, in the province of West Java, has a population of approximately 2.4 million inhabitants. Much of the diagnosis and treatment of TB is carried out in Community Health Centres (CHCs) and public hospitals. Diagnosis by sputum microscopy and provision of anti-TB medication in these health facilities is provided free of charge by the National TB Program (NTP). Approximately 2200 sputum smear-positive TB patients are notified per year from the 80 CHCs located throughout the city (~28 TB patients per CHC), each of which have a defined catchment area.

We present a protocol of a study aimed to measure the TB prevalence and incidence in households and neighbourhoods of a known TB case. We also aim to assess how genomic and epidemiological relatedness varies between TB cases from household contacts, and those from the neighbourhoods.

# Methods

## Study design

The study will interlink with an ongoing randomised trial of an intervention to increase notifications of TB cases from private practitioners (PP) [16]. Thirty-six CHC areas in Bandung have been randomised 1:1 to receive a multi-component intervention with PPs or continue the standard practice. Our study will recruit participants from six of the intervention areas included in the trial selected to be in proximity to each other. The study consists of two main components: 1) a prospective cohort study; and 2) whole-genome sequencing (WGS) and social network analyses (SNA) (Fig 1).

**1) Prospective cohort study.** *Study participation and recruitment.* From 250 bacteriologically-proven TB patients (Index case) living within one of the six selected CHC areas of Bandung City (as described above), we will conduct active case finding among household contacts

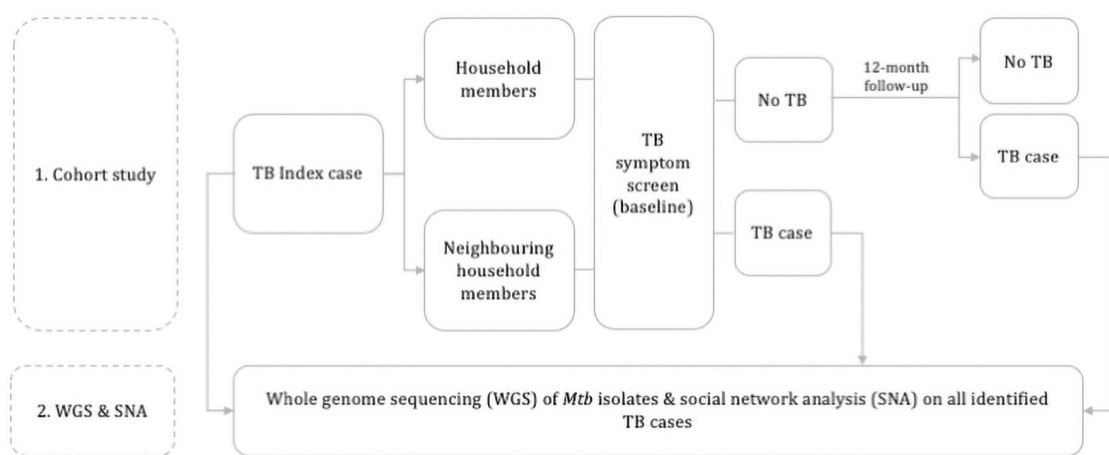

**Fig 1. Overall study schematic.**

(approximately four people per household) and neighbourhood members. Trained research staff will interview all persons aged 10 years and above living in the same household as the Index case, as well as all those living in the five houses closest to the Index case. We will record all subjects in the target households and assess their eligibility. We will not exclude any person with extra-pulmonary TB or being re-treated for TB for screening or follow-up. We will exclude any person in a household who normally lives at another location, or children under the age of 10 years.

The TB nurse in each recruitment site (CHC or hospital) will identify potential Index cases, ensure they are resident in the selected area and seek consent for their contact details to be forwarded for the study. Research staff will then contact consenting Index cases to arrange a time to visit them and their household when most household members are likely to be present. At the visit, further information will be provided and written informed consent obtained. The Index case will also be informed that we wish to contact neighbourhood households and their consent sought before proceeding. Research staff will identify five houses that are the closest in proximity from the Index case house. Within each identified neighbourhood household, the same procedure of informed consent will be followed. The research staff will be instructed not to reveal the identity of the Index case. Consent for any household members aged between 10 and 18 years will be sought from their parent or guardian.

*Data collection and screening algorithm.* For each Index case, socio-demographic information and results from their Acid-fast bacilli (AFB) smear and/or Xpert MTB/RIF testing at the CHC or hospital will be obtained. Sputum samples will be sent to the research laboratory for culture and storage. From each consenting household contact a symptom screen will be undertaken (Fig 2) as well as enquiry about any history of TB, whether they are on any TB preventive therapy, have any other immunocompromised medical condition, and their socio-demographic information (age, gender, ethnicity, smoking status). The location of each household will be recorded using an electronic geographic coordinate system. Any person with symptoms suggestive of TB (weight loss, night sweats, fever, chest discomfort or pain) will undergo clinical assessment. Any person with a productive cough of any duration will be asked to provide two sputum samples for AFB smear and/or Xpert MTB/RIF testing, and *Mtb* culture [17, 18]. All participants will be offered a chest x-ray, followed by sputum examination if the x-ray has changes suggestive of TB. Chest x-ray reports will be read by a radiologist. Cases with a

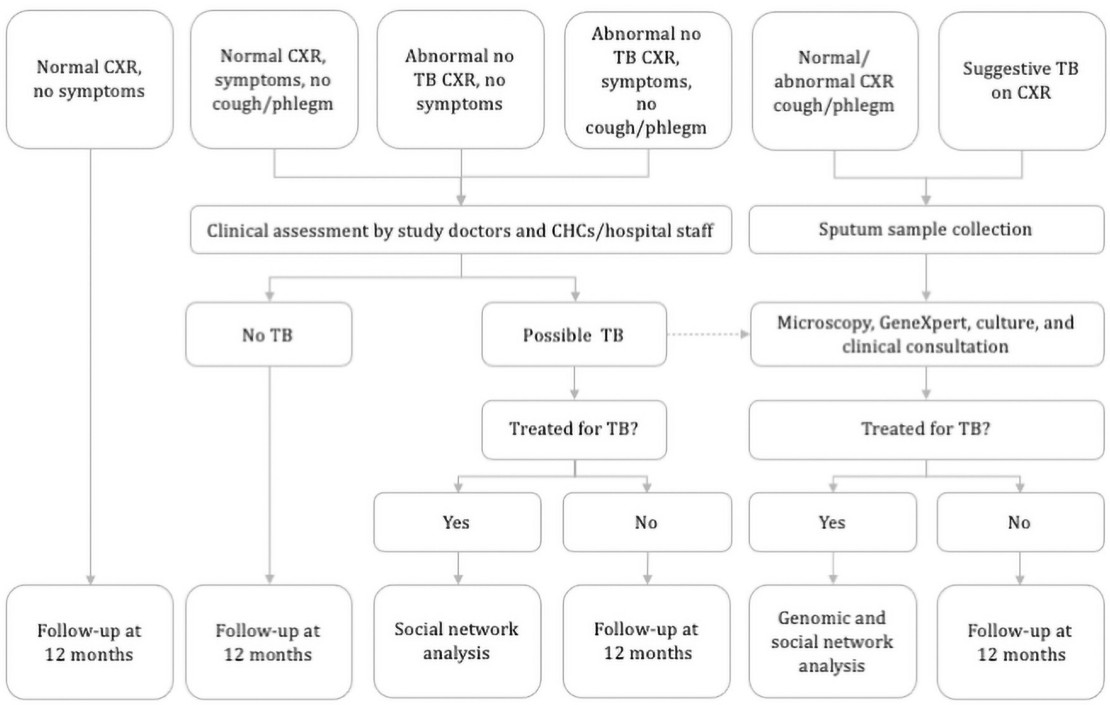

**Fig 2. Baseline screening algorithm.**

radiology report showing indication of TB without bacteriology positivity will be consulted to an infectious disease specialist to determine whether they should receive TB treatment or not. All diagnostic procedures and treatment for any person found to have TB will be undertaken according to National TB guidelines.

*Follow-up.* All participants with no evidence of TB at baseline will be contacted by phone approximately four-monthly to ask whether they have been diagnosed with TB in the interim. Also, to enable the study team to capture the sputum sample and obtain a culture at the time of diagnosis, household contacts will be given a referral card with the study contact details they can show to any CHC or hospital should they present with any TB symptoms or undergo any TB related diagnostic tests. At 12 months a repeat in-person symptom screen will be done (Fig 3). If an individual within a household leaves the neighbourhood area and cannot be followed-up, the approximate date they left will be ascertained from other household members. If a whole household leaves the area before the 12-month follow-up and are unable to be contacted, an approximate date will be confirmed from neighbours.

**2) Whole-genome sequencing and social network analysis.** From each household contact who is diagnosed with TB, as well as Index cases, an interview will be undertaken using a social network questionnaire, and sputum cultured for Mtb will undergo WGS to 'fine map' their relatedness [19]. We will use the BWA-MEM algorithm to map the sequence data to the H37Rv reference genome. Variants will be called using Pilon (v1.22). We will assign a call as missing if the valid depth at a specific site is less than 12, if the mean read mapping quality at the site does not reach seven, or if none of the alternative alleles account for at least 85% of the valid coverage.

Pairwise genetic distances between each TB patient will be determined, after removing single-nucleotide polymorphisms (SNPs) in highly repetitive genes such as the PE/PPE genes and

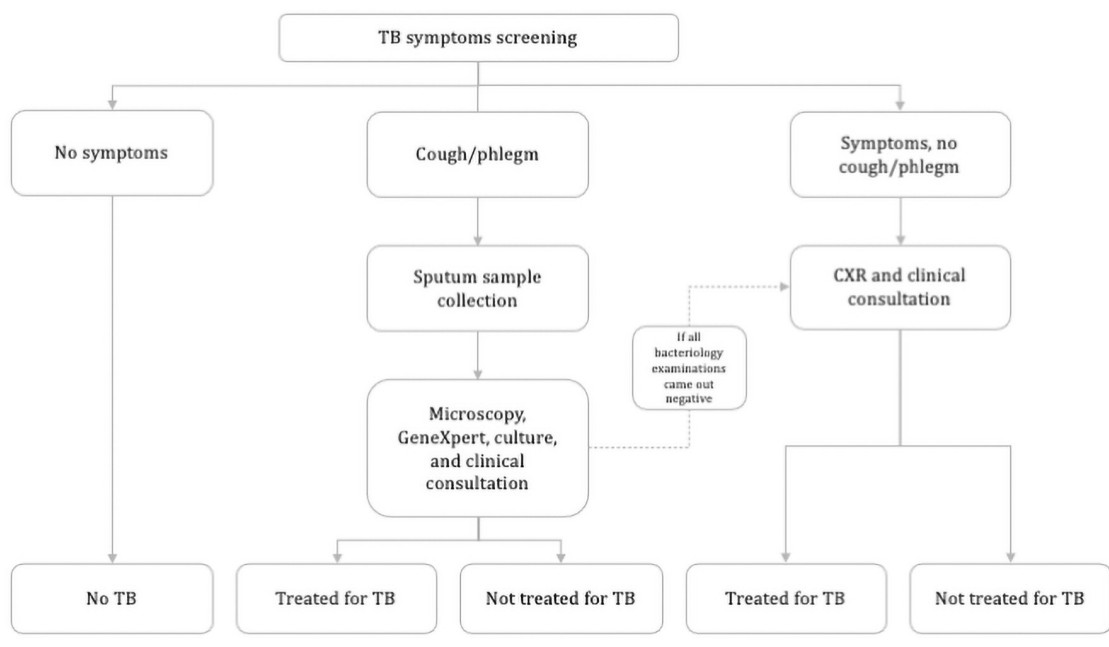

**Fig 3. Follow-up screening algorithm.**

loci under strong selection. [20, 21]. For all pairs, we will obtain the shortest travel time between two points as an approximation of the geographic distance using Open Source Routing Machine, a publicly available tool that analyses Open Street Map road network data. We will assess the relationship between geographic and genetic distance across all pairwise samples to determine the relative contribution of close versus distant contacts in TB transmission. For five definitions of SNPs distances (0–1, 2–3, 4–5, 6–10, and 11–30 SNPs different), we will first compare the odds of genetic relatedness at each of the five levels among pairs within the household, pairs within the neighbourhood, and pairs from different neighbourhoods. We will repeat the same but classifying the pairs in to five categories (within the household, 0–5, 5–10, 10–20, and 20 or more minutes of travel time apart). We will also determine whether any of the household contacts share common meeting places such as employment venues, places of worship, markets, or restaurants.

## Sample size

**Prevalence in neighbourhoods.**   The community estimate of TB prevalence from the Indonesian National Prevalence Survey result for Java was 0.5% (95% CI 0.4 to 0.6%) [22]. We expect a higher rate among TB neighbourhoods, and pragmatically (to justify a policy change), a prevalence of approximately 1.25% culture positive TB cases will be needed in the neighbourhoods. With 4500 neighbourhood participants, the 95% confidence interval around 1.25% will be 0.9 to 1.6%, comfortably above the community estimate.

**Genomic relatedness.**   We expect a maximum of 10% (95% CI 6.3–13.7%) of 250 community diagnosed Index TB cases will have an *Mtb* isolate that is genomically 'identical' to another Index case, compared to approximately 60% of Index TB case/household contact TB case pairs. Percentages for Index case/neighbourhood case pairs will lie between these values. Assuming we identify 35 culture-positive cases in contacts in 250 households and 75 culture-

positive cases in neighbourhoods, we will have >95% power to detect significant differences (p = 0.05 level of significance), in percent genomically 'identical', for comparisons between community and Index TB case/household TB case pairs and between community and Index TB case/neighbourhood case pairs. We will also have over 80% power to detect a significant difference for a household versus neighbourhood comparison, if 60% or more of Index case/household case pairs are genomically 'identical' and 30% or less of Index case/neighbourhood case pairs are genomically 'identical'.

## Outcome measures and statistical analysis

We will categorize TB for Index and household/neighbourhood contacts into four categories: definite TB, probable TB, possible TB, and not TB. Subjects are categorised as definite TB if their *Mtb* culture or Xpert MTB/RIF positive; probable TB if they have TB symptoms, chest x-ray suggestive of TB, and AFB smear-positive but culture negative/not done; possible TB if they have TB symptoms and/or chest x-ray suggestive of TB but all bacteriology examinations negative/not done; and not TB if there was no evidence of TB on a symptom review or investigations. Chest x-rays will be classified as normal, suggestive active TB, suggestive inactive TB, or abnormal not TB based on the radiology report.

The key outcomes of this study are to estimate the TB prevalence and incidence in neighbourhoods around known TB cases, compared to households of known TB cases. TB prevalence defined as the number, percent and 95% confidence intervals (CI) of positive TB cases, according to each TB definition (definite, probable, possible), will be estimated. These proportions will be compared between households, neighbourhoods, and estimates from the Indonesian National Prevalence Survey for Java/Bali. TB incidence defined as incidence rates and 95% CI of positive TB cases according to each TB definition (definite, probable, possible), will be calculated per 1000 person-days. Person-days will be calculated as the days from the date of baseline examination to either: a) the date of TB diagnosis, b) date of death (if died) as reported by household member, c) date of 12-month follow-up examination, or d) date left the neighbourhood area for any household member who has left the area before the 12-month follow-up.

## Data handling

Information collected during interviews will be recorded on an electronic case report form (CRF) or paper-based CRF in case of internet connection problem. Standard operating procedures on interview procedure and data entry will provide the necessary guidelines for research staff. A monthly check will be done for data completeness and accuracy. All databases and data collection tablets will be password protected.

## Ethical considerations

**Research ethics approval.** The protocol and consent forms have been reviewed and approved by the University of Otago Human Ethics Committee (reference number H20-054) on 28 May 2020, translated into Bahasa Indonesia and approved by the Universitas Padjadjaran Health Research Committee (reference number 1129/UN6.KEP/EC/2020) on 3 December 2020.

**Confidentiality.** Participants will be assured of their confidentiality, that no identifiable information will be shared with anyone outside of the research team, and that they can refuse to answer any question if they feel uncomfortable. Each study participant will be given a unique study identification number that will be on all study documentation and in the database. The names and contact details of participants will be required in order to contact them

for the 12-month follow-up interview, but these details will only be known to the relevant research nurse. The database with identifiable names and addresses will be separate from the main database that is available to researchers for the analysis. Signed consent forms will be stored in a locked cabinet.

**Potential risks and benefits.** TB continues to be a stigmatising disease and therefore Index cases may feel exposed and not wish to be identified. This risk will be minimised by the Index case first being asked by CHC staff if they are willing for their name and contact details to be given to the research staff. At the visit, Index cases will be asked for their consent to screen their household members, as well as neighbouring household members. They will be given reassurance that their identity and their TB status will not be revealed to neighbouring households and staff will be trained in issues of confidentiality and follow standard operating procedures. All participants will be informed about the study and given the opportunity to discuss it with their families before giving their consent. COVID-19 continues to be prevalent in Indonesia, therefore strict safety protocols will be adhered to by the research team to prevent the risk of transmission.

## Discussion

The World Health Organization (WHO) 2020 Global TB Report showed Indonesia had the second highest TB incidence in absolute numbers [1]. In 2015, the National TB Program budget was estimated to be 133 million US dollars, much of which is unfunded [15]. This study will confirm whether neighbourhoods in the vicinity of routinely diagnosed TB index cases constitute a high-risk sub-population that may warrant active intervention to enhance TB control. Evidence provided will be crucial for the Indonesian TB Control Program in scaling up efforts towards the elimination of TB in a targeted and cost-effective way.

The combination of high precision genotyping, expanded social network questionnaires, and network analyses provides the opportunity to explore *Mtb* transmission. Genomic and social network analyses will provide important information to guide the design of possible interventions that may include active case finding, preventive treatment, and/or enhanced infection control measures in specific locations. For example, if 'recent' transmission of *Mtb* from index case households to neighbourhoods is indicated as explaining a large component of high neighbourhood prevalence, an intensive intervention of active case finding for early diagnosis and treatment would be warranted. If transmission 'hotspots' within the neighbourhoods can be identified, infection control measures could be considered. If high prevalence and incidence of TB is confirmed but there is a relatively low rate of genomic match within neighbourhoods, it may be that socioeconomic factors that promote reactivation are predominant. An intervention of screening for latent TB infection and initiation of preventive treatment to reduce the possibility of reactivation could be considered, or the focus could be on socioeconomic factors themselves, as suggested by WHO as part of the global strategy to End TB [23].

The proportion of TB cases that can be included in the analysis will be crucial to the ability to interpret these data. As COVID-19 continues to be prevalent in Indonesia, it is possible that recruitment and follow-up will be hindered with fewer than expected numbers being included in the study. This information will be closely monitored and every attempt made to follow-up participants through regular phone contact.

## Conclusions

The study results will provide crucial information to guide the design of possible active case finding interventions to enhance TB control in Indonesia.

## Acknowledgments

We would like to acknowledge Yudi Mulyana Hidayat, the Dean of Faculty of Medicine, Universitas Padjadjaran, and Ahyani Raksanegara, the Head of Bandung City Health Office for their support to carry out this project.

## Author Contributions

**Conceptualization:** Philip Campbell Hill, Susan Margaret McAllister.

**Funding acquisition:** Bachti Alisjahbana, Bony Wiem Lestari.

**Investigation:** Bachti Alisjahbana, Raspati Cundarani Koesoemadinata, Bony Wiem Lestari.

**Methodology:** Bachti Alisjahbana, Chuan-Chin Huang, Megan Murray.

**Project administration:** Raspati Cundarani Koesoemadinata, Panji Fortuna Hadisoemarto, Bony Wiem Lestari, Sri Hartati, Lidya Chaidir.

**Supervision:** Bachti Alisjahbana, Philip Campbell Hill, Susan Margaret McAllister.

**Validation:** Raspati Cundarani Koesoemadinata.

**Writing – original draft:** Bachti Alisjahbana, Raspati Cundarani Koesoemadinata, Susan Margaret McAllister.

**Writing – review & editing:** Bony Wiem Lestari, Chuan-Chin Huang, Megan Murray, Philip Campbell Hill.

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
