## [Decision Letter · Decision Letter 0]

25 Jun 2021

PONE-D-21-15501

Are neighborhoods of tuberculosis cases a high-risk population for active intervention? A protocol for tuberculosis active case finding in Bandung, Indonesia

PLOS ONE

Dear Dr. Koesoemadinata,

Thank you for submitting your manuscript to PLOS ONE. After careful consideration, we feel that it has merit but does not fully meet PLOS ONE’s publication criteria as it currently stands. Therefore, we invite you to submit a revised version of the manuscript that addresses the points raised during the review process.

We look forward to receiving your revised manuscript.

Kind regards,

Seyed Ehtesham Hasnain

Academic Editor

PLOS ONE

Journal Requirements:

1. Please ensure that your manuscript meets PLOS ONE's style requirements, including those for file naming. The PLOS ONE style templates can be found athttps://journals.plos.org/plosone/s/file?id=wjVg/PLOSOne_formatting_sample_main_body.pdf and https://journals.plos.org/plosone/s/file?id=ba62/PLOSOne_formatting_sample_title_authors_affiliations.pdf

2. Please upload a copy of Figure 2 and 3, to which you refer in your text on pages 6 and 7. If the figure is no longer to be included as part of the submission please remove all reference to it within the text.

Additional Editor Comments (if provided):

Minor revision

Reviewers' comments:

Reviewer's Responses to Questions

**Comments to the Author**

1. Does the manuscript provide a valid rationale for the proposed study, with clearly identified and justified research questions?

Reviewer #1: Yes

Reviewer #2: Yes

2. Is the protocol technically sound and planned in a manner that will lead to a meaningful outcome and allow testing the stated hypotheses?

Reviewer #1: Yes

Reviewer #2: Yes

3. Is the methodology feasible and described in sufficient detail to allow the work to be replicable?

Reviewer #1: Yes

Reviewer #2: Yes

4. Have the authors described where all data underlying the findings will be made available when the study is complete?

Reviewer #1: Yes

Reviewer #2: Yes

5. Is the manuscript presented in an intelligible fashion and written in standard English?

Reviewer #1: Yes

Reviewer #2: Yes

6. Review Comments to the Author

You may also provide optional suggestions and comments to authors that they might find helpful in planning their study.

Reviewer #1: The authors have worked on the protocol in finding the active tuberculosis cases in their neighborhoods of Bandung from Indonesia. The study measures the TB prevalence and incidence in households and neighborhoods in the vicinity of known TB cases, and to assess how genomic and epidemiological relatedness varies between TB cases from household contacts, and those from the neighborhoods. The manuscript is well designed; statistics and other analyses are performed to a high technical standard and are described in sufficient detail. Conclusions needs to be supported by the data.

Comments

1. As you have not excluded the comorbidity, did you studied what is the prevalence and incidence rate in households or neighborhoods, who have comorbidity.

2. According to prevalence neighborhoods have higher incidence rate than households, whereas genomical identity is lower in neighborhoods than households. Can you explain these and how did you find these differences?

3. What is the percentage of drop outs during the follow up in your study ?

4. No data has been generated or analyzed in the current study. You have mentioned SNA analysis, it will be more informative if you can add those images.

5. Discussion and conclusion of the study is not informative, it is poorly structured due to lack of previous studies or citations in the neighborhood. It is better to rewrite to understand easily.

6. Correct the grammatical mistakes in the manuscript.

.

Reviewer #2: Comments:

Presented study protocol may help in the findings to measure TB prevalence and incidence in household contacts as well as neighborhoods in the surrounding area of known TB cases from our society. It has been written very nicely, however few advices may improve this protocol:

• While collecting the data of socio-demographic information, author may take collect the common TB comorbidities include diabetes mellitus, smoking, alcohol-use disorders, chronic lung diseases, cancer and depression data also.

• In patient history, alcoholism and smoker status of patient as well as neighborhoods contact should be noted if possible. Because these behavior are strongly associated with TB disease and attributed to high risk of SARs in elderly and TB comorbidities patients.

• It is also advised to perform phenotypic DST to all culture positive specimens in addition of genotypic DST.

• EPTB is already excluded from this study. However, it is also advised to make a separate group for them to analyzed data considering all methodology as discussed.

Conclusion: Authors are advised to reframe.

7. PLOS authors have the option to publish the peer review history of their article (what does this mean?). If published, this will include your full peer review and any attached files.

Reviewer #1: No

Reviewer #2: No

---

## [Author Response · Author response to Decision Letter 0]

22 Jul 2021

22 July 2021

Seyed Ehtesham Hasnain

Academic Editor

PLOS ONE

Dear Dr Hasnain,

Re: Manuscript PONE-D-21-15501: “Are neighbourhoods of tuberculosis cases a high-risk population for active intervention? A protocol for tuberculosis active case finding in Bandung, Indonesia”

Thank you for review of the above-stated manuscript and the opportunity to submit a revised version. 

The two reviewers made helpful suggestions for improvement which we believe we have addressed in the revised paper or in our rebuttal below. We have responded to each point raised by the reviewers and have updated the manuscript, indicated as track changes.

Thank you for your consideration

Kind regards

Raspati C. Koesoemadinata 

(Corresponding author)

Response to Reviewers

Journal Requirements:

Thank you. We have revised the manuscript, and we believe it complies with PLOS ONE's style requirements.

2. Please upload a copy of Figure 2 and 3, to which you refer in your text on pages 6 and 7. If the figure is no longer to be included as part of the submission please remove all reference to it within the text.

Figures 2 and 3 were uploaded with the original submission. We will upload them again with this revised submission to make sure they are accessible to the reviewers.

We uploaded the research grant review and ethical approval letters and mistakenly labelled them as "Supporting information". They are now labelled as “Other”.

The reference list has been checked. No changes have been made to the reference list. 

Reviewer’s Comments to the Author

Reviewer #1: The authors have worked on the protocol in finding the active tuberculosis cases in their neighbourhoods of Bandung from Indonesia. The study measures the TB prevalence and incidence in households and neighbourhoods in the vicinity of known TB cases, and to assess how genomic and epidemiological relatedness varies between TB cases from household contacts, and those from the neighbourhoods. The manuscript is well designed; statistics and other analyses are performed to a high technical standard and are described in sufficient detail. Conclusions needs to be supported by the data.

Comments

1. As you have not excluded the comorbidity, did you studied what is the prevalence and incidence rate in households or neighbourhoods, who have comorbidity.

We will ask the participants about any other immunocompromised medical conditions. This information has now been added to the manuscript (lines 129-130).

2. According to prevalence neighbourhoods have higher incidence rate than households, whereas genomical identity is lower in neighbourhoods than households. Can you explain these and how did you find these differences?

In the earlier study in Bandung, we have found a higher TB incidence rate in the neighbourhoods. This study has shown us that the incidence rate of self-reported TB was 649 per 100,000/year in TB household contacts, 675 in neighbouring households, and 325 in a randomly selected community (Reference #8). Evidence from other studies also have reported the higher TB incidence in the community outside TB households (Reference #5) (Lines 54-55). We are unsure of the genomic identity of cases in households and neighbourhoods, hence the need for this study. 

3. What is the percentage of drop outs during the follow up in your study?

Several measures have been put in place to minimise drop-out, such as regular phone calls to participants throughout the 12 months, having phone numbers of other household members (Lines 146-156). The estimated percentage of drop-out that may occur during the follow-up period is not known particularly during this time of the COVID-19 pandemic. This limitation has been added to the discussion (lines 300-303). 

4. No data has been generated or analyzed in the current study. You have mentioned SNA analysis, it will be more informative if you can add those images.

Our manuscript is a protocol paper. Recruitment of participants has only recently started. Full results, including SNA analysis figures will be included in the future manuscripts. 

5. Discussion and conclusion of the study is not informative; it is poorly structured due to lack of previous studies or citations in the neighbourhood. It is better to rewrite to understand easily.

As our manuscript is a protocol paper, it is impossible to format it according to the standard formatting of a discussion with results. We believe we have included information required by the journal in discussing the possible implications of our study and how the results might be used. Extra information has been added on limitations. 

6. Correct the grammatical mistakes in the manuscript.

Thank you. We have reviewed the manuscript thoroughly and corrected the mistakes.

Reviewer #2: Comments:

Presented study protocol may help in the findings to measure TB prevalence and incidence in household contacts as well as neighbourhoods in the surrounding area of known TB cases from our society. It has been written very nicely; however few advices may improve this protocol:

1. While collecting the data of socio-demographic information, author may take collect the common TB comorbidities include diabetes mellitus, smoking, alcohol-use disorders, chronic lung diseases, cancer and depression data also.

Thank you. Yes, questions about comorbidities, other TB risk related conditions including smoking status, are included in the questionnaire. This information has now been added to the manuscript for clarity (lines 129-130). We have found a very low prevalence of alcohol use (<2%) in our previous study therefore questions about alcohol use were not included in the current study. 

2. In patient history, alcoholism and smoker status of patient as well as neighbourhoods contact should be noted if possible. Because these behavior are strongly associated with TB disease and attributed to high risk of SARs in elderly and TB comorbidities patients.

Thank you. Please see the above response

3. It is also advised to perform phenotypic DST to all culture positive specimens in addition of genotypic DST.

We don’t consider that phenotypic DST is necessary for this study and its objectives. Moreover, it would increase the costs of the study beyond the budget allocated. Therefore, we will not include it in our study. 

4. EPTB is already excluded from this study. However, it is also advised to make a separate group for them to analyzed data considering all methodology as discussed.

Thank you for your suggestion. Yes, we will not include EPTB patients as the index case. However, we will record all EPTB cases found during the household or the neighbourhood screening and report them in the results (Lines 105-106).

Conclusion: Authors are advised to reframe.

It is unclear what the reviewer is asking. As this is a protocol paper, a conclusion is not a requirement according to the manuscript guidelines.

---

## [Editor Report · Decision Letter 1]

29 Jul 2021

Are neighbourhoods of tuberculosis cases a high-risk population for active intervention? A protocol for tuberculosis active case finding

PONE-D-21-15501R1

Dear Dr. Koesoemadinata,

We’re pleased to inform you that your manuscript has been judged scientifically suitable for publication and will be formally accepted for publication once it meets all outstanding technical requirements.

Kind regards,

Seyed Ehtesham Hasnain

Academic Editor

PLOS ONE

Additional Editor Comments (optional):

I have gone through the revised manuscript and also the Authors response to the comments of the reviewers. The manuscript was sent for revision and Authors have modified the manuscript keeping in mind the comments of the Reviewers. The important issue of co-morbidity has been addressed along with dropouts and incidence rates. Their response about phenotypic DST is acceptable.All grammatical and typographic errors have been taken care off by the Authors. In my view, the authors have otherwise satisfactorily addressed all the comments made by the reviewers and added all required information, and have revised the manuscript accordingly. I recommend this manuscript for publication.
---

## [Editor Report · Acceptance letter]

5 Aug 2021

PONE-D-21-15501R1 

Are neighbourhoods of tuberculosis cases a high-risk population for active intervention? A protocol for tuberculosis active case finding 

Dear Dr. Koesoemadinata:

I'm pleased to inform you that your manuscript has been deemed suitable for publication in PLOS ONE. Congratulations! Your manuscript is now with our production department. 

Kind regards, 

on behalf of

Prof. Seyed Ehtesham Hasnain 

Academic Editor

PLOS ONE